# Crevice Corrosion Behavior of 201 Stainless Steel in NaCl Solutions with Different pH Values by In Situ Monitoring

**DOI:** 10.3390/ma17051158

**Published:** 2024-03-01

**Authors:** Zejie Zhu, Hang Zhang, Yihan Bai, Pan Liu, Haoran Yuan, Jiangying Wang, Fahe Cao

**Affiliations:** 1School of Materials and Chemistry, China Jiliang University, Hangzhou 310018, China; zhanghcjlu@163.com (H.Z.); baiyhcjlu@163.com (Y.B.); wangjiangying@cjlu.edu.cn (J.W.); 2Frontier Research Initiative, New Industry Creation Hatchery Center (NICHe), Tohoku University, Sendai 980-8579, Japan; 3School of Materials, Sun Yat-Sen University, Shenzhen 518107, China; caofh5@mail.sysu.edu.cn

**Keywords:** crevice corrosion, SECM, stainless steel, pH

## Abstract

Crevice corrosion (CC) behavior of 201 stainless steel (SS) in 1 M NaCl + x M HCl/y M NaOH solutions with various pH was investigated using SECM and optical microscopic observations. Results show that the CC was initiated by the decrease in pH value within the crevice. The pH value near the crevice mouth falls rapidly to 1.38 in the first 2 h in the strongly acidic solution, while the pH value was observed to rise firstly and then decrease in the neutral and alkaline solutions. It indicates there is no incubation phase in the CC evolution of 201-SS in a pH = 2.00 solution, while an incubation phase was observed in pH = 7.00 and 11.00 solutions. Additionally, there appeared to be a radial pH variation within the gap over time. The pH value is the lowest at the gap mouth, which is in line with the in situ optical observation result that the severely corroded region is at the mouth of the gap. The decrease in pH value inside results in the negative shift of open circuit potential (OCP) and the initiation of CC of 201-SS. The increased anodic dissolution rate in the acidic solution accelerates the breakdown of passive film inside, reducing the initiation time and stimulating the spread of CC.

## 1. Introduction

Stainless steel (SS) has been extensively utilized in harsh working environments including nuclear plants, petrochemical, and maritime buildings due to its superior mechanical properties and good corrosion resistance [1,2,3,4]. For instance, SSs are usually used in nuclear power facilities, steam generator tubes, and coolant pump casings. The corrosion rate of SS is a crucial factor in determining the lifespan and safety of equipment. Crevice corrosion (CC) is one of the most common corrosion phenomena; crevice positions such as conduit connections, gaskets below flanged joints, beneath washers and bolt heads are vulnerable to CC [5,6,7]. Typically, CC is very subtle and has a relatively long induction time. The material structure may be distinctly damaged when the CC phenomenon is observed [8]. Therefore, it is of significance to elaborate the corrosion process and intrinsic mechanism of CC.

Currently, there are two primary theories about CC [3,5,9]: the critical crevice solution theory (CCST) and the IR drop theory (IRRT). As per CCST, changes of chemical composition inside the crevice (such as a decrease in pH value and an increase in Cl^−^ concentration) could break the passive film [10,11,12,13], which consequently accelerates the generation of CC. In addition, based on the IR drop theory (“I” represents the current flowing through the corrosion circuits, while “R” refers to the solution resistance in the crevice), the IR drop is the crucial factor inducing the occurrence of CC [14]. CC would be initiated when the potential inside is in the active zone of the polarization curve. The IR drop is evidently influenced by various factors such as solution conductivity, crevice size, and chemical composition within the crevice [15,16,17]. The potential window of active corrosion has been observed to extend due to the fluctuation in chemical composition in the occluded gap [18]. However, some researchers believe that neither CCST nor IRRT solely could explain the mechanism of CC. Generally, pit creation and proliferation are the initial causes of the CC, and the propagation stage of CC is influenced by both the solution environment and IR drop. According to previous research, aggressive ions are generated and trapped in the fissure during the initial pitting period, which changes the chemical environment inside; then, the passive film would be broken and continuous active dissolution of metal takes place. These theories indicate the chemical composition environment plays an important role in the initiation and evolution of CC in a narrow gap. These compositional variations are mainly affected by the pitting and diffusion process of the chemical species inside the fissure. Additionally, there exist disparities in the local radial dissolution behavior of metals within the gap across various systems. The corrosion reactions taking place inside the gap can lead to alterations in the regional chemical composition of the immediate surrounding environment, thereby significantly influencing the local corrosion process of CC. Consequently, it is imperative to gather real-time data on the radial chemical environmental variations within the crevice for a thorough understanding of the corrosion processes.

The evolution of CC is influenced by numerous factors, including the chemical environment (the pH and dissolved oxygen) [19,20], temperature [21], the type of electrolyte [22], applied stress [23], and crevice size, etc. [24,25]. Jakobsen [26] discovered that a higher temperature can facilitate the dissolving of steel, making CC easier to be initiated. Standish also studied the effect of temperature on the propagation of the CC process of titanium alloy; the results showed that the active dissolution of metal inside the crevice proceeds more rapidly at a higher temperature. It has also been reported that the addition of trace H_2_S into the electrolyte can reduce the CC rate of carbon steel, which is attributed to a thin sulfide layer developed on the steel surface [22]. Zhu et al. [6] found increased stress improves the likelihood that metastable pits change to stable pits, which then promotes the development of CC. Chen et al. [27] investigated the influence of crevice geometry on the corrosion behavior of 304-SS; their study demonstrated that different crevice width or length would affect the dissolved oxygen concentration, which would lead to various oxidation behaviors in the fissure. The bulk solution environment is another important factor which could affect the corrosion’s evolution. Mu et al. [28] found X70 steel could suffer from CC in pH = 2 and pH = 12 solutions, while CC could not be initiated in a pH = 6.8 solution. It was also discovered that carbon steel often exhibits passivation behavior in an alkaline solution, and active dissolution in an acidic or neutral solution. However, little work has been done to study the effect of bulk-solution acidity on the local chemical composition and electrochemical behavior within the crevice. It is challenging to build an in situ corrosion monitoring system and obtain real-time radial composition data in a crevice, because the large electrode size is insufficient for high-resolution monitoring of the ion dynamics in the micro-zone inside the fissure. Thus, it is quite challenging to assess the effect of the pH value of the bulk solution on the regional corrosion behaviors in the metal crevice; hence, the development of in situ micro-region detection methods has become quite important [29,30,31,32].

Nowadays, more and more researchers pay attention to SECM [33,34,35,36,37,38,39] due to its high spatial resolution and versatility. The ultramicroelectrode (UME) (in micro-scale or nano-scale) combined with a 3D mobile platform permits it to capture micro morphological and chemical data in situ over the interface [40,41,42,43,44,45]. In our earlier research, a submicron Pt/IrO*_x_*-pH UME was applied to monitor the local radial pH distribution and evolution within the gap, using the potentiometric mode of SECM [41]. However, the impact of solution pH value on the mechanism of localized corrosion behavior of 201-SS and its correlation with radial pH distribution within the crevice has not been extensively studied.

In this work, the local CC behavior of 201-SS in 1 M NaCl solutions with various pH values was investigated using SECM and optical microscopic observations. The goal is to evaluate the impact of the pH value of bulk solution on the initiation and evolution process of CC and its correlation with radial pH distribution inside crevice, based on a self-designed high-resolution submicron UME combined with SECM potentiometric mode.

## 2. Materials and Methods

### 2.1. Material and Solution Preparation

Commercial 201-SS was utilized as a specimen in this research. The 201-SS samples used in this work were purchased from Foshan Hongwang Stainless Steel Co., Ltd. (Foshan, China). The chemical composition of the specimen (wt.%) is shown in Table 1. The specimen’s surface was meticulously ground using SiC grinding paper with 800, 1000, and 2000 grit in succession. Then it was polished to a mirror sheen using 1 μm silica suspension, and cleaned ultrasonically with ethanol and deionized water. CC tests were conducted in 1 M NaCl with various pH values (pH = 2.00 (1 M NaCl + 0.01 M HCl), 7.00 (1 M NaCl), and 11.00 (1 M NaCl + 1 mM NaOH), respectively). The pH value of the solution was adjusted using HCl or NaOH solution. All experiments were conducted at room temperature.

### 2.2. CC Test

The self-assembly device for CC experiments is exhibited in Figure 1. Full details of the setup were described in our previous study [41]. A glass plate and a polytetrafluoroethylene (PTFE) gasket were used to create a 100 μm gap. Threaded stainless-steel bolts and nuts were sealed with hot glue. The exposed specimen areas inside and outside were 4 × 7 and 5 × 6 mm, respectively. Prior to immersion, the gap was filled with electrolyte via a micro-syringe. There were four holes on the glass plate: one hole was used to place the inner reference electrode, and the other three holes (∅ = 100 μm) along a line were used for pH measuring in SECM tests. A submicron pH (Pt/IrO*_x_* − pH) UME was created (with a shape-controllable diameter of approximately 1 μm) as working electrode to investigate the local radial pH distribution and changes within the gap by the application of SECM’s potentiometric mode. The fabrication procedure of the pH UME was described in our earlier research [41]. Additionally, Pt wire was used as the counter electrode. The setup and procedure for traditional electrochemical analysis and SECM tests are described in detail in the Appendix A. Unless otherwise specified, all potentials are relative to the potential of the Ag/AgCl reference electrode (0.206 V vs. standard hydrogen electrode at 24.85 °C).

### 2.3. Corrosion Morphology Observation

A metallographic optical microscope (OM, M230-HD228S, AOSW, Changzhou City, China) was used to monitor the morphology of crevice samples in situ at different positions. After 48 h immersion, the specimens were removed from the experimental apparatus. Deionized water was subsequently used to clean the surface of the samples. Before surface analysis, the specimens were stored in a desiccator to lessen the risk of electrode damage. The composition and content information of the corrosion products were analyzed via a scanning electron microscope (SEM, JSM-7001F, JEOL, Akishima, Japan, 15 kV-Mapping, BSD Full) with energy dispersive X-ray spectroscopy (EDS, 15 kV-Map) and a Fourier transform infrared spectrometer (FTIR, TENSOR27, Brucker, Karlsruhe, Germany) with the infrared wavelength ranging from 320 to 4000 cm^−1^.

## 3. Results

### 3.1. Effect of Bulk Solution pH Value on OCP

Figure 2 depicts the evolution of OCP of 201-SS samples in 1 M NaCl solutions with various pH values. The OCP value moves in the positive direction as the pH increases. It is acknowledged that passive film could remain intact in the alkaline solution. Therefore, it is deduced that the sample is less prone to CC as pH increases. Moreover, the OCP in the pH = 2.00 solution is lowest and remains stable during the test, which is attributed to the constant instability of passive film in a strongly acidic environment. The metal is in an active dissolution state during the immersion. In the solutions of pH = 7.00 and 11.00, the samples exhibit a similar variation tendency: as the soaking time increases, the OCP changes gradually to a negative potential, which is owing to the change of chemical composition within the fissure. With the prolongation of test time, the cathodic reactants are consumed and CC occurs inside the gap. Some metal cations released from the metal dissolution would hydrolyze, resulting in a pH drop inside. Consequently, the passive film inside gradually becomes unstable, leading to the negative change of OCP with time. The OCP value of the sample in the pH = 11.00 solution decreases less than that in the pH = 7.00 solution, which may be owing to the passive film being more stable in an alkaline solution.

### 3.2. Effect of Bulk Solution pH Value on Corrosion Potentials

Figure 3 depicts the polarization curve of the 201-SS sample measured in 1 M NaCl solutions with various pH values. Full details about fitting values of the corrosion potentials and current are listed in Table 2.

As shown, the shape of the three polarization curves is quite similar. An identifiable passivation zone can be seen for all samples. The corrosion current (*i_corr_*) and passive current increase as the pH value of the bulk solution decreases. The corrosion potential (*E_corr_*) of the sample decreases with the decrease in the pH value of the bulk solution. These results indicate that the 201-SS in the alkaline solution shows relatively better corrosion resistance. These findings agree well with the OCP result.

### 3.3. Effect of Bulk Solution pH Value on the Radial pH Distribution Variation Inside

Figure 4 illustrates the evolution of radial pH distribution inside in 1 M NaCl solutions with various pH values. It could be seen that the pH value at different positions is apparently different: the pH value shifts positively as the distance increases from the crevice mouth. This is because the anodic dissolution mainly occurred near the gap mouth, as the hydrolysis reaction of metal cations leads to a pH drop at the crevice opening. This conclusion is in line with our earlier finding [41]. The variation of pH value with time inside reflects the development process of corrosion. The different pH evolution lines at the crevice mouth indicate the various rates of CC initiation and propagation.

In the strongly acidic (pH = 2.00, Figure 4a) solution, the pH value at the opening falls rapidly to 1.38 in the first 2 h and reaches a steady state from 2 to 8 h, then rapidly declines to 1.13 after 8 h. According to the pH variation characteristics, three CC periods can be distinguished: the activation phase, the stable development phase, and the rapid development phase. During the activation period, some metals begin to dissolve, causing the pH of the solution to decrease, which subsequently accelerates the dissolution of the metals. After 2 h, CC turns into the stable development phase, and the pH keeps relatively stable. After 8 h, CC is in the rapid development phase. A mass of metal dissolution reactions takes place, resulting in a sharp pH drop. Figure 4b shows the pH evolution in the pH = 7.00 solution. As the figure shows, four phases could be identified. From the beginning to 1 h, CC is in the incubation phase. The cathodic process occurs at the opening. The reduction of O_2_ or H^+^ leads to a slight rise in pH value. From 1 h to 4 h, the pH value declines quickly, and CC evolves into the activation phase. From 4 to 8 h, the pH variation is relatively small, which indicates the stable development of CC. After 8 h, the pH value drops suddenly, which means CC reaches the rapid development phase. In the pH = 11.00 solution (Figure 4c), CC undergoes three stages: the incubation phase, the activation phase, and the rapid development phase, in which the CC process is similar to that described in the pH = 7.00 solution.

Additionally, it can be observed that the trends of pH value change at the middle and bottom of the crevice remain consistent in each pH solution: it grows steadily from 0.5 to 1 h, and then drops as time goes on. This is because the passive film at the crevice center and bottom is intact in the first 1 h. The reduction of O_2_ or H^+^ in the blocked space results in a gradual increase in pH value. After 1 h, the diffusion of protons from the crevice opening to the bottom leads to a drop in pH value.

### 3.4. In Situ CC Morphologies Observation

Figure 5 illustrates the in situ CC micrographs at different positions of the samples in 1 M NaCl solutions with various pH values. As it shows, the pit growth processes inside of the crevice varied in different acidic solutions.

Figure 5A shows the in situ CC micrographs in the pH = 2.00 solution. It is visible that a few small black spots and bubbles appear and grow from the beginning to 2 h in the vicinity of the gap mouth. It suggests pitting corrosion takes place immediately near the opening at the beginning of the test. The metal dissolves directly at the crevice opening. The hydrogen evolution reaction in a strongly acidic solution is responsible for the emergence of bubbles. From 2 to 8 h, the number of spots increases steadily with time, which is in line with the minor pH fluctuation in Figure 4a. A few small black spots appear at the middle of the crevice, which means pitting corrosion takes place in the middle area of the gap at this period. After 8 h, it can be seen that a substantial amount of corrosion products accumulates at the opening and extends to the inner part of the crevice, indicating the CC enters into the rapid development phase.

Figure 5B shows the in situ CC micrographs in the pH = 7.00 solution. From 0.5 to 1 h, no discernible morphological alteration is observed inside, indicating the CC is still in the incubation phase. During this period, the passive film remains intact, and homogeneous corrosion occurs both inside and outside the fissure. From 1 to 4 h, no discernible morphological alteration is observed at the opening. However, the pH value near the mouth decreases with time during this period, according to the pH variation in Figure 4b. It suggests some metastable pits appear during the activation phase. From 4 to 8 h, some black spots appear and grow steadily, which means the CC moves into the stable development phase. After 8 h, a substantial amount of corrosion products accumulates at the opening, which means CC is in the rapid development phase.

The evolution of CC process of 201-SS in the pH = 11.00 solution is shown in Figure 5C. From 0.5 to 1 h, there are no obvious morphological changes inside, which means the CC is still in the incubation phase, which is owing to the stability of passive film in an alkaline solution. Similarly, from 1 to 8 h, CC steps into the activation stage. No discernible morphological alteration is observed at the opening, while the pH value declines during this period, according to Figure 4c. It suggests metastable pits appear. Some dissolved metal ions hydrolyze and the pH decreases. After 8 h, the area of the corrosion region inside in the pH = 11.00 solution is less than that in the pH = 7.00 solution, indicating the passive film is more stable in an alkaline solution.

### 3.5. Ex Situ Observation of the Morphologies and Corrosion Products

Figure 6 shows the ex situ surface corrosion morphology images of 201-SS after 48 h immersion in different pH solutions. As seen, the most serious CC occurs in the pH = 2.00 solution. Large amounts of corrosion rusts accumulate at the gap mouth. While the 201-SS in the pH = 7.00 solution suffers a moderate CC attack, the sample in the pH = 11.00 solution is only slightly eroded, which is in agreement with the in situ optical microscopic observation results.

Figure 7 displays the SEM images and elements analysis results of the corrosion products in different pH solutions. The results show that the products contain Fe, Cr, O, and Cl elements, which suggests the presence of iron (or chromium) hydroxides or oxides in the corrosion products. The appearance of the Cl element in corrosion products is owing to adsorption of sodium chloride electrolyte or the creation of unstable iron oxychloride during the test. Additionally, the EDX signal intensity of O element is highest in the pH = 2.00 solution, suggesting more corrosion products accumulated on the steel surface in the strongly acidic solution.

The FTIR spectra of the corrosion product composition are depicted in Figure 8. Five bands at 3853, 3427, 1632, 1044, and 580 cm^−1^ are present for corrosion products of all samples in different pH solutions, indicating the composition of corrosion products is identical. As shown, a faint absorption signal at around 3853 cm^−1^ is attributed to the presence of the Fe-OH-Fe group, which arises from the hydrolysis of iron ions; it indicates the generation of the metal hydroxide. The band located at 1632 cm^−1^ is the water molecules’ O-H bending band. The frequency of 1044 cm^−1^ designates the Cl-O bending vibration peak, suggesting the formation of iron oxychloride inside the crack. The broad IR band at about 580 cm^−1^ can be assigned to the bending vibration peak of the Cr-O bond. This shows that chromium-containing oxide is synthesized. The FTIR spectra clearly demonstrate the formation of metal hydroxides (Fe(OH)_2_ and Cr(OH)_3_) and iron oxychloride on the 201-SS sample surface, which is in agreement with the EDS data.

## 4. Discussion

Based on the above-mentioned results, it was proposed that the acidity of the bulk solution exerts a significant impact on the CC behavior of 201-SS. According to Figure 4 and Figure 5, it is shown that metastable pits firstly appear and grow near the crevice mouth. A large number of cations released from the initial pits hydrolyze, resulting in a pH drop inside. When pH becomes low enough to destroy the passive film within the crack, more pits appear and lead to a sustained pH drop. Therefore, the initiation and spreading process of CC is correlated with the passivation state of metal and the pH value inside.

### 4.1. The CC Mechanism of 201-SS in pH = 2.00 Solution

The CC process of 201-SS in the pH = 2.00 solution is divided into three stages: the activation, stable development, and rapid development phases. Figure 9 shows the CC process schematic of 201-SS in the pH = 2.00 solution.

The activation phase refers to the period from the beginning to 2 h. A few small black spots and bubbles appear and grow with time in the vicinity of the gap mouth (Figure 5A). It means pitting corrosion occurs near the crevice mouth immediately at the beginning. This is possibly owing to the instability of the passive film in a strongly acidic solution. The appearance of bubbles is due to the hydrogen evolution reaction in a highly acidic solution.
(1)$$\mathrm{Fe}\to {\mathrm{Fe}}^{2+}+2e^{-}$$
(2)$${\mathrm{Fe}}^{2+}+2H_{2}O\to \mathrm{Fe}{\left(\mathrm{OH}\right)}_{2}↓+2H^{+}$$
(3)$$\mathrm{Cr}\to {\mathrm{Cr}}^{3+}+3e^{-}$$
(4)$${\mathrm{Cr}}^{3+}+3H_{2}O\to \mathrm{Cr}{\left(\mathrm{OH}\right)}_{3}↓+3H^{+}$$

During this time, there are no discernible morphological changes at the middle or bottom of the gap (Figure 5A), and the pH value at the middle or bottom grows steadily from 0.5 to 1 h, then drops as time goes on (Figure 4a). This is because the concentration of cathode reactant inside is equal to that outside. O_2_ or H^+^ is consumed (as per the following reactions) at the middle and bottom of crevice in the initial 1 h.
(5)$$O_{2}+H_{2}O+4e^{-}\to 4{\mathrm{OH}}^{-}$$
(6)$$2H^{+}+2e^{-}\to H_{2}↑$$

After 1 h, there is a diffusion of protons from the crevice opening to the interior of the crevice. From 2 to 8 h, CC of 201-SS moves into the stable development phase. The number of spots increases with time steadily at the opening (Figure 5A), and the pH fluctuation is small at the opening (Figure 4a). The pH decreases steadily at the middle and bottom of the fissure. This is owing to the hydrolysis of metal ions and the diffusion of protons from the crevice opening. Meanwhile, the concentration of Cl^−^ would rise inside due to the immigration from the bulk solution to the gap solution. The environment inside becomes more aggressive, which facilitates the growth of metastable pits. After 8 h, CC steps into the rapid development phase. This phase is characterized by the accumulation of massive corrosion products and a rapid pH drop inside. The corroded region extends to the inside of the crack. This signals the solution concentration of corrosive reactants is approaching the critical point, which speeds up the metal’s disintegration (or corrosion current) rate and raises the potential drop. When the IR drop hits a crucial value, the location and intensity of corrosion attack become potential-dependent.

### 4.2. The CC Mechanism of 201-SS in pH = 7.00 and 11.00 Solutions

The initiation and development of the CC process in the pH = 7.00 and 11.00 solutions are different from that in the pH = 2.00 solution. The CC process of 201-SS in the pH = 7.00 solution includes four stages: the incubation, activation, stable development, and rapid development phases. Figure 10 displays the schematic plot of the CC process in the pH = 7.00 solution. From the beginning to 1 h, CC is in the incubation phase. No discernible morphological alterations are observed inside, while the pH value inside rises gradually during this stage (Figure 4b). The reduction of oxygen or protons within the gap results in a slight pH rise. From 1 to 4 h, CC enters into the activation phase, metastable pits appear inside and some metal ions released from the dissolution hydrolyze, leading to the pH decrease in Figure 4b, which facilitates the growth of metastable pits. From 4 to 8 h, CC is in the stable development phase. Some black spots appear and increase steadily. The pH value fluctuates less near the opening. Pitting corrosion occurs in this phase. Some chloride ions would immigrate from the bulk solution to the gap solution. The corrosion current increases continuously. After 8 h, CC is in the rapid development phase. A substantial amount of metal dissolves at the opening, and the corroded region extends to the opening of the crack.

Figure 11 shows the evolution of CC of 201-SS in the pH = 11.00 solution. There are three phases in the CC process of 201-SS in a pH = 11.00 solution. The first two phases (incubation phase and activation phase) are the same as the process in the pH = 7.00 solution. After 8 h, CC steps into the rapid development phase, and a substantial amount of corrosion products accumulates at the opening (Figure 11c), which is consistent with the result in Figure 4c. In this phase, the concentration of corrosive reactants approaches the critical point, which speeds up the metal’s disintegration (or corrosion current) rate and raises the potential drop. The location and intensity of corrosion attack become potential-dependent.

From the discussion above, it can be inferred that the initiation of CC of 201-SS cannot be well explained solely by the CCST or IRRT. The initial sign of CC appears in the form of metastable pits. The metal near the crevice opening suffers the most severe corrosion attack. This result agrees with our earlier research [41]. The incubation and activation phases of CC were significantly influenced by the local pH variation.

The evolution of CC in various pH solutions is different. There is no incubation phase in the pH = 2.00 solution. The corrosion rate of 201-SS in the pH = 2.00 solution is faster than that in the pH = 7.00 and 11.00 solutions, as the passive film is brittle and prone to rupture in a strongly acidic solution. While there is an incubation phase of the CC in solutions with pH values of 7.00 and 11.00, the cathodic and anodic reactions take place both outside and inside the fissure uniformly. As the CC proceeds, metastable pits appear inside. Metal cations are produced and the concentration of Cl^−^ would rise inside due to the immigration from the bulk solution to the gap solution. When the concentration of these aggressive ions arrives at the critical level, the passive film inside would be completely broken. The dissolution rate of metal inside increases. The corrosion rate of 201-SS is smallest in the pH = 11.00 solution, which may be explained by the fact that the passive film of metal usually remains intact in an alkaline solution.

## 5. Conclusions

CC behaviors of 201-SS in 1 M NaCl solutions with different pH values are investigated in this study. The above experimental data show that the bulk solution acidity exerts a significant impact on the CC behavior of 201-SS. The primary conclusions are as follows:(1)The sample is most seriously corroded in the pH = 2.00 solution, while it is scarcely damaged in the pH = 11.00 solution. The internal erosion area grows with the drop of pH value;(2)There is a clear radial pH distribution along the crevice wall. The biggest pH drop is observed at the opening, followed by the middle and the bottom areas;(3)The evolution of CC of 201-SS in various pH solutions is different. There is no incubation phase in the pH = 2.00 solution. In contrast, an incubation phase is observed in the CC development of 201-SS in the pH = 7.00 and 11.00 solutions;(4)The decrease in pH value inside results in a negative shift of OCP and the initiation of CC of 201-SS. The increased anodic dissolution rate in the acidic solution accelerates the breakdown of passive film within the gap and stimulates the spread of crevice corrosion;(5)The CC mechanism of 201-SS in various pH solutions can be well described by the combination of CCST and IRRT. The initiation and spreading process of CC is correlated with the passivation state of metal and the pH value inside.

## Figures and Tables

**Figure 1 materials-17-01158-f001:**
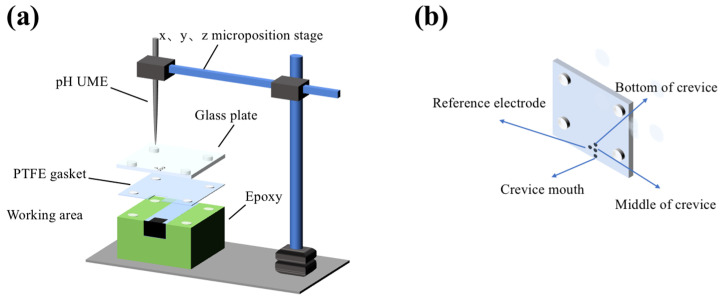
(**a**) The schematic diagram of the self-assembly device used for CC studies and (**b**) the detailed image of the glass plate.

**Figure 2 materials-17-01158-f002:**
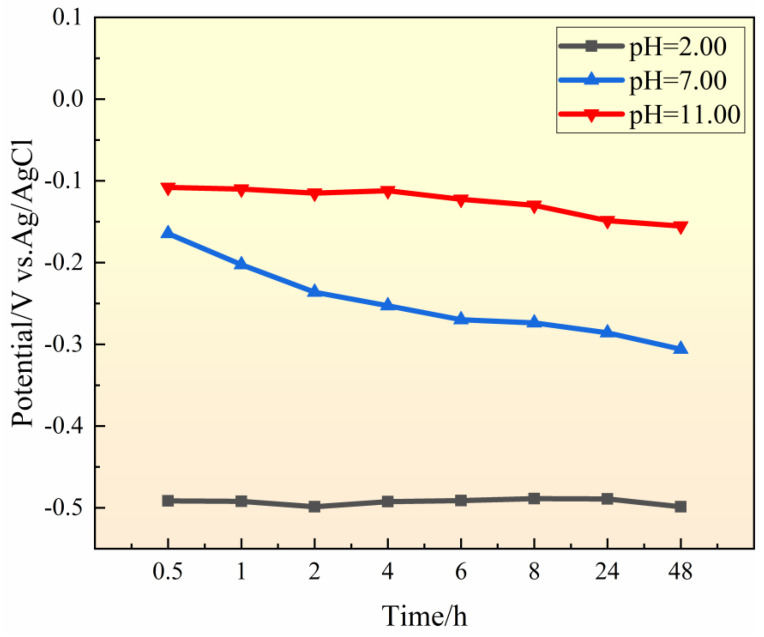
Time dependence of OCP of 201-SS in 1 M NaCl solutions with different pH values at room temperature.

**Figure 3 materials-17-01158-f003:**
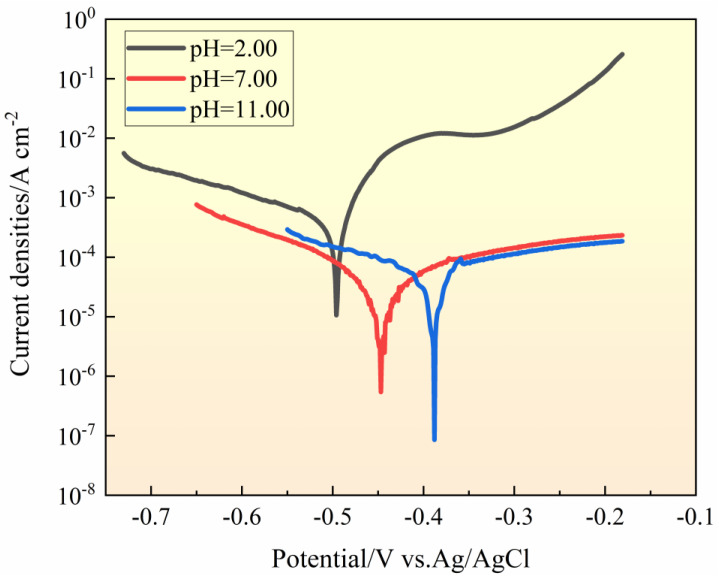
Polarization curves of 201-SS measured in 1 M NaCl solutions with different pH values at room temperature. Scan rate is 1 mV/s.

**Figure 4 materials-17-01158-f004:**
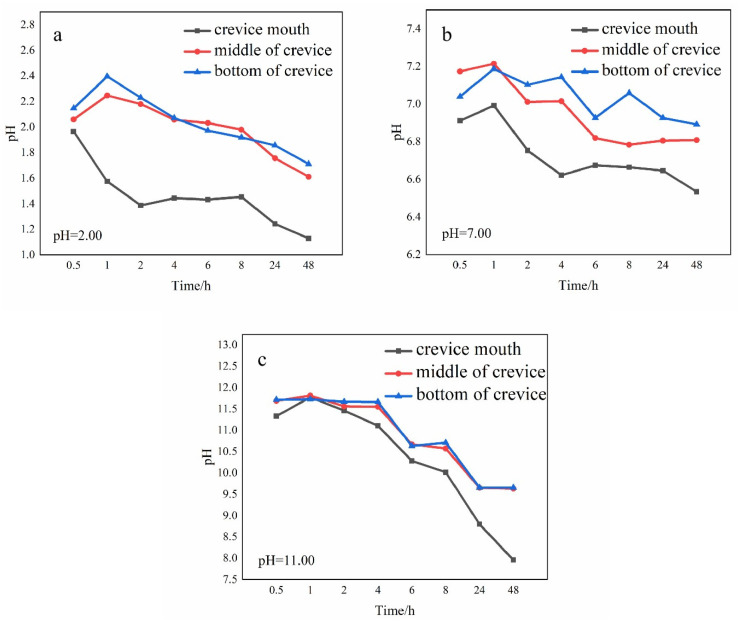
Evolution of the radial pH value within the 201-SS crevice in 1 M NaCl bulk solution with pH values of (**a**) 2.00, (**b**) 7.00, and (**c**) 11.00, respectively.

**Figure 5 materials-17-01158-f005:**
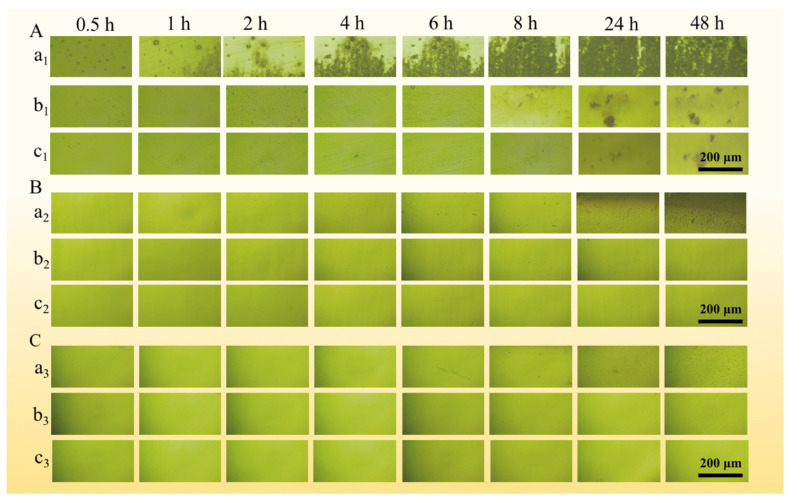
In situ optical observations of corrosion evolution of different positions in the crevice of 201-SS samples in 1 M NaCl bulk solutions with pH values of (**A**) 2.00, (**B**) 7.00, and (**C**) 11.00 at room temperature. Radial morphology inside the crevice: (a_1_–a_3_) Crevice mouth, (b_1_–b_3_) crevice middle, and (c_1_–c_3_) crevice bottom.

**Figure 6 materials-17-01158-f006:**
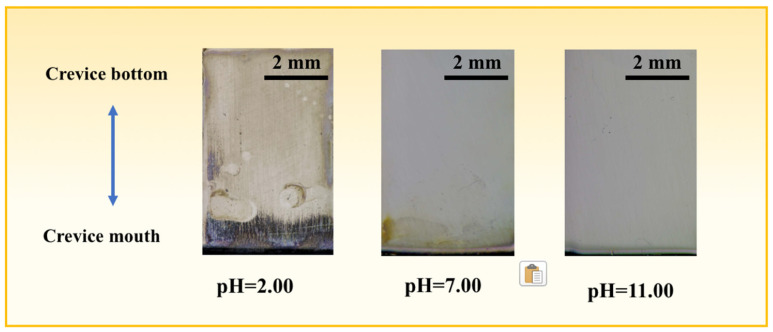
Observations of the corrosion morphology of 201-SS after being immersed in 1 M NaCl solutions with pH of 2.00, 7.00, and 11.00 at room temperature for 48 h.

**Figure 7 materials-17-01158-f007:**
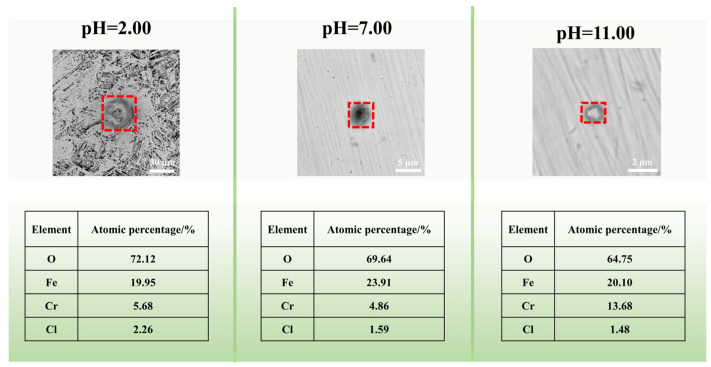
SEM images of the corrosion products of 201-SS after being immersed in 1 M NaCl solutions with different pH values for 48 h at room temperature as well as the elemental analysis of the target areas. The region enclosed by the red framework represents corrosion products.

**Figure 8 materials-17-01158-f008:**
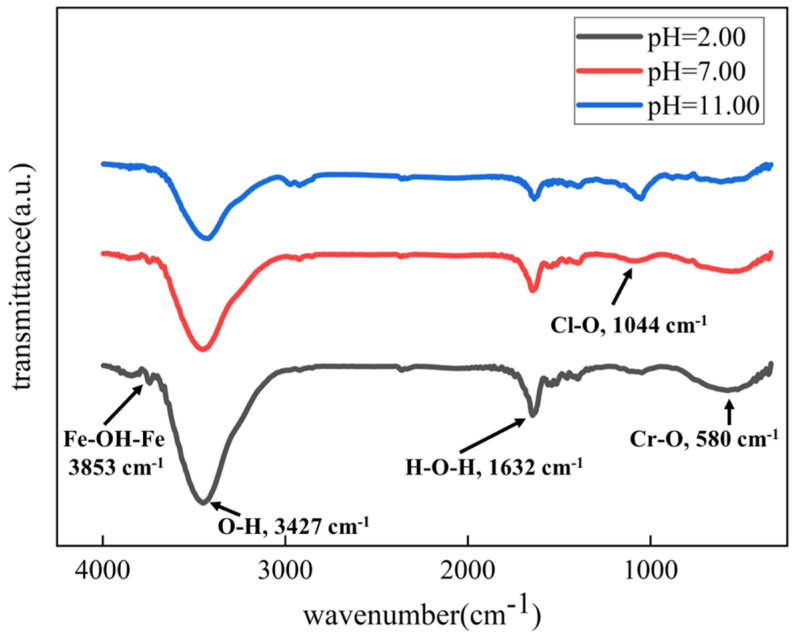
Infrared spectroscopy of the corrosion products around the crevice mouth of the 201-SS sample after being immersed in 1 M NaCl solutions with various pH values at room temperature for 48 h.

**Figure 9 materials-17-01158-f009:**
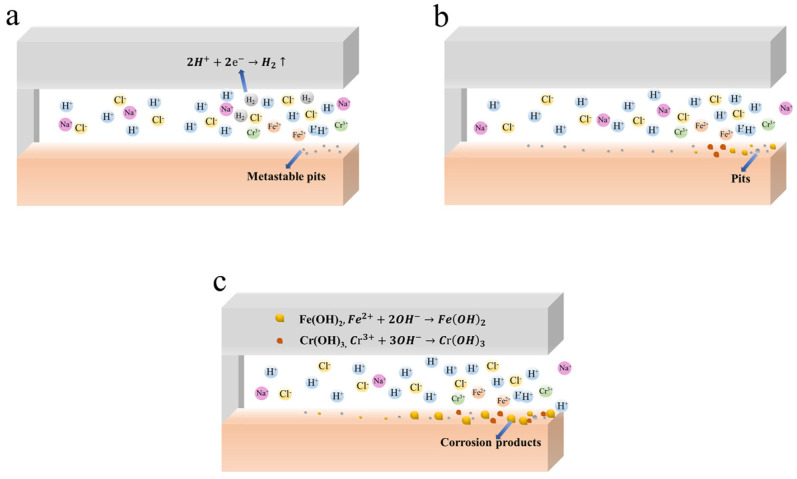
Schematic diagram of the CC process of 201-SS in the pH = 2.00 solution: (**a**) Activation phase, (**b**) Stable development phase, (**c**) Rapid development phase.

**Figure 10 materials-17-01158-f010:**
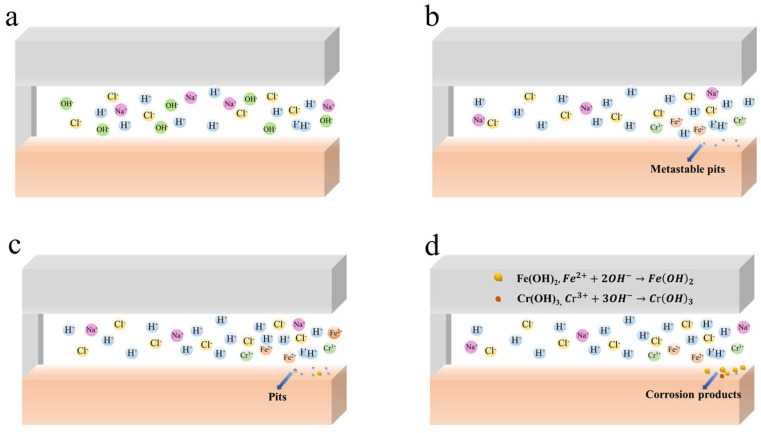
Schematic diagram of the CC process of 201-SS in the pH = 7.00 solution: (**a**) Incubation phase, (**b**) Activation phase, (**c**) Stable development phase, (**d**) Rapid development phase.

**Figure 11 materials-17-01158-f011:**
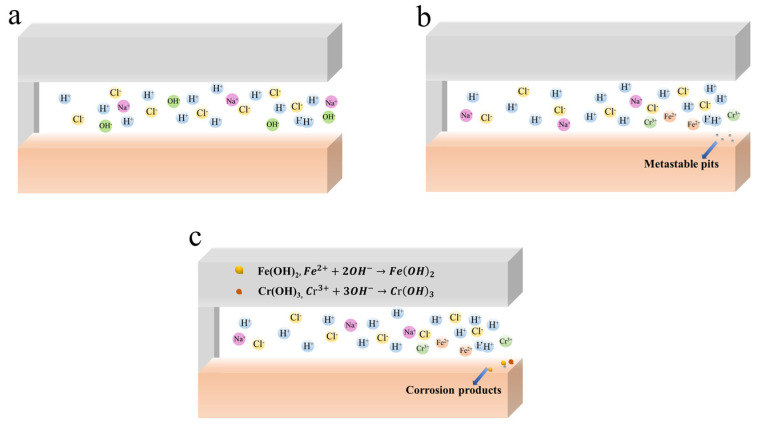
Schematic diagram of the CC process of 201-SS in the pH = 11.00 solution: (**a**) Incubation phase, (**b**) Activation phase, (**c**) Rapid development phase.

**Table 1 materials-17-01158-t001:** The chemical composition of the specimen (wt.%).

Elements	Fe	Cr	Ni	Mn	Si	C	P	S
Content	77.24	14.09	5.25	1.49	1.43	0.25	0.13	0.12

**Table 2 materials-17-01158-t002:** The fitting values of corrosion potential (*E_corr_*) and current (*i_corr_*) of 201-SS in 1 M NaCl solutions with different pH values.

pH	2.00	7.00	11.00
*E_corr_*/V	−0.5257	−0.4432	−0.3872
*i_corr_*/A·cm^2^	7.68 × 10^−4^	4.52 × 10^−5^	2.44 × 10^−5^

## Data Availability

The data presented in this study are available on request from the corresponding author.

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
