# Peer review of "Crevice Corrosion Behavior of 201 Stainless Steel in NaCl Solutions with Different pH Values by In Situ Monitoring"

_materials, 2024, doi:10.3390/ma17051158_

Round 1

Reviewer 1 Report

Comments and Suggestions for Authors The manuscript investigates the onset and development of crevice corrosion on SS 201 in NaCl. The results are very noble because they question the two existing theories on crevice corrosion, namely CCST and IRRT theory. To support the assumptions and conclusions, sophisticated methods are applied. I do not have any major remarks that could improve the manuscript. However, I want to raise the concern that the authors have published a similar article in Electrochimica Acta (ref. 41 in the manuscript). In order to avoid self-plagiarism, please give detailed explanation what are the differences between a) diagrams on the evolution of the radial pH value within the 201-SS crevice over time; b) optical images of corrosion evolution of different positions in the crevice of 201-286 SS samples; c) FTIR spectra and conclusions on the corrosion product composition in this manuscript and ref. 41; since these results are presented in both manuscripts. Comments on the Quality of English Language Minor language comments: please rephrase the text: "CCST thinks the alteration in the chemistry of solution... " "While IRRT believes the IR drop is crucial in the beginning of CC... "

Reviewer 2 Report

Comments and Suggestions for Authors

Title- Crevice corrosion behaviour of 201 stainless steel in NaCl solution with different pH values by in-situ monitoring

Authors- Z. Zgu, H. Zhang, Y. Bai, P. Liu, H. Yuan, J. Wang, F. Cao

Manuscript Id- Materials-2845139

In this manuscript, the authors have investigated crevice corrosion of SS201 in different pH electrolytes. Although the work is not new, it has the potential to contribute some important information to the corrosion community. After a careful review with interest, I found that the manuscript falls short on the discussion part. The discussion presented in the manuscript are mare speculations of the fact. The authors are also advised to polish the manuscript on the grounds of the English language to meet the quality of scientific writing and so is the standards of MDPI Materials.

Below are some comments/suggestions for the authors;

[1]  The authors should consider rewriting the abstract part. Written in a more generic way and lacks scientific depth. Please rephrase the sentences. It is nice to add some factual data in the abstract for better readability. For ex., how much % increase or decrease in CC rates and so on? The authors should also specify which “initial phase”. Any specific name?

[2]  Again, in the introduction, the information presented is very generic. Can the authors please specify what extra/new information they are adding (for the better understanding of readers)?

[3]  Novelty is missing in the introduction. Crevice corrosion in SS is not a new concept. Furthermore, investing CC via SECM should not be a novelty. All in all, I didn’t find any scientific merit of this work. Please define what extra/new results have been added that will significantly help the corrosion community.

[4]  How the SS201 is fabricated? Which processing route? Is it fabricated in-house or borrowed from somewhere else? The authors should mention it.

[5]  If HCl and NaOH are added to control the pH then it would be completely a different electrolyte. So, in what sense the authors are arguing they performed CC tests in 1M NaCl? Instead, it would be appropriate to state (1M NaCl + xM HCl) and (1M NaCl + yM NaOH).

[6]  Again, what is the rationale behind using different pH? Not new. Already reported in the literature.

[7]  The mechanism which authors argue “passive film formed in alkaline solution remain intact” is unclear. It would be appreciated if the authors could provide more insights into the same.

[8]  The reviewer is curious about the change in corrosion morphology under different pH. The morphology is changing from localized intense corrosion (in acid) to localized pits? It would be nice if the authors could analyze the pit size in each sample and present it in the form of a critical pit size where the corrosion attack is shifting.

[9]  Furthermore, the reviewer is also wondering about the white colour particle in Fig. 7c (pH=11). Is it the residue of NaCl? Please confirm with EDS once.

[10]     A quick question: why the elemental composition of the corroded surface in each sample is same?

[11]     Where did the authors see metastable passivation in Tafel plots? I doubt even the passivation is not happening during the experiments. Any comments?

Comments on the Quality of English Language

Needs moderate improvement.

Reviewer 3 Report

Comments and Suggestions for Authors

The paper presents the crevice corrosion behaviour of 201 stainless steel in NaCl solution. The mechanisms of crevice corrosion initiation in acidic, neutral and basic environments were analysed by SECM and optical microscopy. Concerning the corrosion initiation mechanism, it was found that in neutral or basic pH solutions the passive film remains intact for a period of time, thus there is an initiation phase, unlike crevice corrosion in acidic environments when this phase was not evident. On the other hand, research has revealed that pH varies over time at the gap, being lower at the mouth of the gap, which leads to an increased rate of anodic dissolution in acidic environments, accelerating the degradation of the passive film inside, thus reducing the initiation time and stimulating CC spreading.

The article is of interest of Materials journal, but specific aspects mentioned in the following require the revision of the paper (minor revision).

- The paper should be reworded to be shorter. The information provided is not particularly extensive, so it can be presented in fewer paragraphs.

-On the other hand, some information could be detailed, such as: the exact composition of steel 201 (not catalogue data), as some details could influence corrosion behaviour.

- The experiments were conducted at room temperature. The article does not discuss the influence of other factors on corrosion in crevices (e.g. temperature, type of electrolyte), factors that can influence the corrosion mechanism and definitely influence the degradation rate. Although they are not specifically the subject of the research, these issues could be discussed in the introductory chapter.

- Although from the way of presenting, it is understood how the experiments were carried out. However, some minor aspects are briefly described (e.g. sample preparation and full parameters used for SEM and FTIR), which makes it difficult to strictly replicate the experiments.

Reviewer 4 Report

Comments and Suggestions for Authors

The work by Zhu et al. (Crevice corrosion behavior of 201 stainless steel in NaCl solution with different pH value by in-situ monitoring) reports the crevice corrosion (CC) behavior of 201 stainless steel (SS) in 1 M NaCl solution with various pH value. For this, they employed SECM and optical microscopic methods. The authors provided various characterization and models for the evaluation.

In general, the manuscript is easy to follow and well-organized. I think it provides valuable information to this research field. I recommend the publication of the report after some minor points given below are addressed.

What does IR stand for in the Introduction?

Please add scale bar to Figure 5 and 6.

I noticed some grammatical errors. Please fix them. 

Comments on the Quality of English Language

Moderate editing of English language required.

Round 2

Reviewer 1 Report

Comments and Suggestions for Authors

In my opinion, the authors have answered the questions posed by reviewers, and the manuscript could be published in the present form.

Reviewer 2 Report

Comments and Suggestions for Authors

The reviewer had a detailed look at the revised version of the manuscript and found that the authors have satisfactorily addressed all the comments raised during the previous round. I am happy to recommend this manuscript for the publication in Materials. Congratulations to all the authors.

Comments on the Quality of English Language

The authors should proof read the manuscript before the publication so as to avoid any grammatical/typo errors.